# Survival after Lung Metastasectomy from Urothelial Carcinoma: A Multi-Institutional Database Study

**DOI:** 10.3390/cancers16193333

**Published:** 2024-09-29

**Authors:** Yoshikane Yamauchi, Masaaki Sato, Takekazu Iwata, Makoto Endo, Norihiko Ikeda, Hiroshi Hashimoto, Tai Hato, Hidemi Suzuki, Haruhisa Matsuguma, Yasushi Shintani, Haruhiko Kondo, Takahiko Oyama, Yoko Azuma, Tomohiko Iida, Noriaki Sakakura, Mingyon Mun, Keisuke Asakura, Takashi Ohtsuka, Hirofumi Uehara, Yukinori Sakao

**Affiliations:** 1Department of Surgery, Teikyo University School of Medicine, Tokyo 173-8605, Japan; 2Department of Thoracic Surgery, The University of Tokyo Graduate School of Medicine, Tokyo 113-8655, Japan; 3Division of Thoracic Surgery, Chiba Cancer Center, Chiba 260-8717, Japan; 4Department of Thoracic Surgery, Yamagata Prefectural Central Hospital, Yamagata 990-2292, Japan; 5Department of Surgery, Tokyo Medical University, Tokyo 160-0023, Japan; 6Department of Thoracic Surgery, National Defense Medical College Hospital, Saitama 359-8513, Japan; 7Department of General Thoracic Surgery, Saitama Medical Center, Kawagoe 350-8550, Japan; 8Department of General Thoracic Surgery, Chiba University Graduate School of Medicine, Chiba 260-8677, Japan; 9Division of Thoracic Surgery, Tochigi Cancer Center, Utsunomiya 320-0834, Japan; 10Department of General Thoracic Surgery, Osaka University Graduate School of Medicine, Osaka 565-0871, Japan; 11Department of Thoracic Surgery and Thyroid Surgery, Kyorin University School of Medicine, Tokyo 181-8611, Japan; 12Department of General Thoracic Surgery, National Hospital Organization Tokyo Medical Center, Tokyo 152-8902, Japan; 13Division of Chest Surgery, Department of Surgery, Toho University School of Medicine, Tokyo 143-8541, Japan; 14Department of Thoracic Surgery, Kimitsu Central Hospital, Kimitsu 292-0822, Japan; 15Department of Thoracic Surgery, Aichi Cancer Center Hospital, Nagoya 464-8681, Japan; 16Department of Thoracic Surgical Oncology, The Cancer Institute Hospital, Tokyo 135-8550, Japan; 17Division of Thoracic Surgery, Department of Surgery, Keio University School of Medicine, Tokyo 160-8582, Japan; 18Division of Thoracic Surgery, Department of Surgery, Jikei University School of Medicine, Tokyo 105-8471, Japan; 19Department of Thoracic Surgery, Hakodate Goryoukaku Hospital, Hakodate 040-8611, Japan

**Keywords:** computed tomography, disease-free survival, multivariate analysis, overall survival, pulmonary metastasectomy, urothelial carcinoma

## Abstract

**Simple Summary:**

This study examined the outcomes of lung metastasectomy in urothelial carcinoma patients using data from a Japanese multi-institutional database. The study included 100 patients who underwent the procedure between 1985 and 2021. Results showed 5-year overall survival and disease-free survival rates of 59% and 46%, respectively. Larger tumor diameter and presence of distant metastases at primary cancer treatment were identified as significant adverse prognostic factors. The authors claim that this is the largest published case series on this topic, providing benchmark data for assessing long-term outcomes.

**Abstract:**

Background/objectives: The efficacy of lung metastasectomy in patients with urothelial carcinoma remains inconclusive, as there is only limited evidence from small studies. In this study, we aimed to assess the prognostic outcomes of excising pulmonary metastases from urothelial carcinoma. Methods: In this study, we utilized data from the Metastatic Lung Tumor Study Group of Japan database, a multi-institutional prospective database of pulmonary metastasectomies. We examined the data of patients who had undergone pulmonary metastasectomy for urothelial carcinoma between 1985 and 2021. Exclusion criteria included insufficient clinical information and follow-up of <3 months. Results: The study cohort comprised 100 patients (63 bladder cancer, 37 renal pelvic and ureteral cancer), with a median follow-up of 34 months. There were 70 male and 30 female patients of average age 66.5 ± 10.4 years at lung metastasectomy. The median interval from treatment of the primary lesion to metastasectomy was 19 months and the maximum tumor diameter was 21 ± 15 mm. Three- and five-year overall survival rates were 69% and 59%, respectively. Three- and five-year disease-free survival rates were 56% and 46%, respectively. Multivariate analysis identified larger tumor diameter (hazard ratio: 1.62, 95% confidence interval: 1.21–2.17) and distant metastases at the time of treatment of the primary cancer (hazard ratio: 4.23; 95% confidence interval: 1.54–11.6) as significant adverse prognostic factors for overall survival. Conclusions: To our knowledge, this is the largest published case series of pulmonary resection for metastatic urothelial carcinoma, providing benchmark data for the assessment of long-term outcomes of this rare entity.

## 1. Introduction

Urothelial carcinoma (UC) is the 10th most prevalent cancer worldwide, ranking second among cancers of the genitourinary system. Between 30 and 50% of patients with localized disease eventually relapse and develop metastases, and 5% present with metastatic urothelial carcinoma (mUC) at diagnosis [1]. The prognosis of mUC is poor, with the 5-year overall survival (OS) being <5% [2]. Recent guidelines recommend systemic chemotherapy for mUC patients [3]; however, the median survival of those undergoing first-line chemotherapy is only 13–15 months [4,5] Alarmingly, 24–61% of mUC patients do not receive chemotherapy because of factors such as diminished performance status and impaired renal function, significantly worsening their prognosis [6,7,8].

Although malignant tumors with lung metastases are generally classified as Stage IV and have poor prognoses, lung metastectomy can be effective in patients who meet specific criteria. In many cancer types, excision of isolated lung metastases in properly selected patients has been shown to improve OS. Thomford et al. proposed the following criteria for considering lung metastasectomy: (1) the patient can tolerate surgery; (2) the primary tumor is controlled; (3) no extrapulmonary metastases; and (4) unilateral lung metastases [9]. These criteria have been widely adopted, sometimes with minor modifications, for determining which lung metastasis patients may benefit from surgical resection.

Pulmonary metastasectomy is a widely accepted treatment for patients with pulmonary metastases [10,11,12]. Resection of metastatic lung tumors has become standard practice in carefully selected patients with various primary malignancies, including colorectal [13,14], uterine [15,16], head and neck [17,18], hepatocellular [19,20], gastric [21], and esophageal cancer [22,23].

Thus, the therapeutic benefit of resection of lung metastases originating from UC warrants examination. Although several groups have reported outcomes of surgery with curative intent in highly selected patients with mUC [24,25,26,27], there are still only limited data on the outcomes and prognosis following resection of such metastases. A few small studies have explored the role of pulmonary resection in this setting [28,29,30]. In the present study, we utilized a database from the Metastatic Lung Tumor Study Group of Japan (METAL-J) to investigate long-term outcomes and factors associated with prolonged survival after pulmonary metastasectomy for mUC.

## 2. Patients and Methods

### 2.1. Database and Ethical Statement

In 1984, the METAL-J established a database of patients who had undergone lung metastectomy. Thus far, 7514 patients have been prospectively registered in this database, all of whom have undergone resection of lung metastases with curative intent. Lung resections performed solely for biopsy purposes were excluded. The database contains the following data: sex, age, primary tumor status, tumor treatment, metastasis status, dates and specifics of metastectomies, disease-free survival (DFS), OS, and other follow-up data.

For this study, clinical and pathological data on patients with pulmonary mUC were collected from the METAL-J database. The data were provided by 22 institutions between October 1985 and December 2021. Only patients who had undergone curative lung metastasectomy were included. Exclusion criteria were (1) unknown prognosis or surgical details, (2) residual tumor in the thoracic cavity, and (3) follow-up <90 days post-surgery. The case selection process is outlined in Figure 1. This retrospective study also received institutional ethics approval (Approval number: 23-102, approval date: 30 October 2023).

Before registration in this database, criteria for pathological diagnosis of the original UC, recurrence of UC, pathological diagnosis of mUC, radiological diagnosis of metastatic lung tumors, and indications for excision of lung tumors varied between participating institutions. Each institution has a dedicated cancer board comprising oncologists, radiologists, pathologists, surgeons, and related specialists who collectively determine all aspects of tumor diagnosis and treatment. In general, metastasectomy aimed to completely remove all pulmonary lesions in patients without evidence of extra-thoracic metastasis or uncontrolled primary tumors on imaging. In patients undergoing lobectomy or segmentectomy, lymph node dissection or sampling was performed if lymph node metastases were present. However, wedge resection was only performed when preoperative positron emission tomography/computed tomography (CT) or other imaging modalities showed no evidence of lymph node involvement. Further, several sequential transurethral bladder tumor resections were defined as a series of local treatments.

### 2.2. Statistical Analysis

Statistical analyses were performed using SPSS version 28 (IBM, Armonk, NY, USA) and GraphPad Prism version 10.2 (GraphPad Prism Software, San Diego, CA, USA) for constructing figures. *p* < 0.05 was considered to denote statistical significance. The optimal cutoff values for continuous prognostic indexes were determined using the method established by Budczies et al. [31], which is described at https://molpathoheidelberg.shinyapps.io/CutoffFinder_v1/ (accessed on 7 September 2024). This method fits Cox proportional hazard models to the dichotomized and survival variables, defining the optimal cutoff as the point with the most significant split. OS was defined as time from pulmonary metastasectomy until death or last follow-up and DFS as time from metastasectomy until further recurrence, death, or last follow-up. Patients alive at last follow-up were censored. Survival curves based on clinicopathological factors were generated via the Kaplan–Meier method and compared using the log-rank test. Univariate and multivariable Cox proportional hazard models were used to assess relationships between clinicopathological factors and survival after metastasectomy, analyzing continuous variables rather than using optimal cutoffs.

## 3. Results

In all, 100 patients with pulmonary metastases from UC were eligible for analysis. Table 1 shows the patients’ clinicopathological characteristics. A total of 70 of them were men and the primary tumor was in the bladder in 63 patients. Nine patients had lymph node metastases and seven distant metastases at the time of initial treatment. In the treatment of the primary tumor, for cases with surgery in which the resection margin could be determined, all were complete resections. In 95 patients the pulmonary metastasis was the first metastasis. Eleven patients had multiple metastases when their pulmonary metastases were detected. Seven patients underwent preoperative chemotherapy. The chemotherapy regimens for these cases were as follows: two cases of dose dense MVAC therapy, two cases of MVAC therapy, one case of gemcitabine plus cisplatin, one case of gemcitabine plus carboplatin, and one case of carboplatin plus paclitaxel. Eleven patients received postoperative chemotherapy. Wedge resection for pulmonary metastasis was performed on 52 patients. There were two cases where tumors remained on the operative side after lung surgery.

Figure 2 shows the OS and DFS after resection of lung mUCs. The 3- and 5-year OS was 69% and 59%, respectively, and the 3- and 5-year DFS was 56% and 46%, respectively.

Table 2 shows the results of univariate analysis of clinicopathological factors for association with OS and DFS after resection of lung mUCs. Analysis of OS identified the following four significant factors: pathologically-proven distant metastases on initial treatment of UC, size of lung metastasis according to CT, size of resected lung metastases, and lymph node metastases found during lung surgery. In contrast, the analysis for DFS identified the following three significant factors: number of lung metastasis detected by CT, preoperative chemotherapy for lung metastasis, and lymph node metastasis found during lung surgery.

Table 3 shows the results of multivariate analysis for OS and DFS after lung resection. Factors were selected according to the results of univariate analyses; however, additional factors for OS included size of lung metastases according to CT and size of resected lung metastases. Because these two factors are so strongly related to each other, we selected only the size of resected lung metastases for this analysis. We selected this factor because the other factor contained many missing values. Multivariate analysis identified pathologically distant metastases on initial treatment of UC and resected tumor size as significant prognostic factors for OS after resection of lung mUC. Additionally, lymph node metastases diagnosed during lung surgery were identified as a significant factor for DFS after resection of lung mUC.

Figure 3 shows Kaplan–Meier curves for the factors identified as significantly associated with prognosis by multivariate analysis. Figure 3a shows two curves for OS after lung metastectomy according to the presence or absence of distant metastases at initial treatment of UC. Patients with distant metastases on initial treatment with UC had a significantly poorer OS than did those without them (*p* = 0.0061). Figure 3b shows two curves for OS according to the size of the resected lung metastases. The optimal cutoff of 22 mm was defined as the point with the most significant split. Patients with tumors larger than 22 mm had a significantly poor OS than did patients with smaller tumors (*p* = 0.0009). Figure 3c shows two curves for DFS according to the presence or absence of lymph node metastases at the time of lung metastectomy. Patients with lymph node metastases at the time of lung surgery had a significantly poorer DFS than those without them (*p* = 0.0035).

## 4. Discussion

Our findings indicate that selected patients can achieve long-term survival after excision of pulmonary metastases from UC. Furthermore, multivariate analyses identified the following factors as associated with poor prognosis: pathologically-proven distant metastasis at the initial treatment for UC, size of lung metastases resected, and presence of lymph node metastases at lung surgery. Although pulmonary metastasectomy is reportedly an effective treatment strategy that achieves long-term survival in patients with various solid tumors, few studies have reported the results of removing pulmonary metastases from UC, possibly because of the extremely poor prognosis once UC has produced distant metastases. The study cohort comprised 100 patients who had undergone pulmonary metastasectomy. To our knowledge, this is the largest cohort reported so far. Thus, the prognostic factors we identified may be more reliable than those reported by smaller studies regarding the outcomes of pulmonary metastasectomy.

At diagnosis, approximately 75% of bladder cancer patients have disease limited to the mucosa (stage Ta, carcinoma in situ) or submucosa (stage T1) [32], with an even higher percentage in patients under 40 years old [33]. Without treatment, over half of carcinoma in situ tumors progress to muscle-invasive disease [34]. However, reliable prognostic factors for clinical decision making are lacking, as many are based on retrospective analyses with limited conclusive power. Advanced age at diagnosis correlates with increased risk of recurrence and progression in pure bladder carcinoma in situ [35]. After recurrence of UC, the prognosis is very poor, with the 5-year survival rate reportedly being approximately 5%. However, previous studies on excision of lung metastases from UC have reported a 5-year OS of 46.5–65.3% and 5-year DFS of 26–37.5% [28,30,36,37], which is consistent with our findings. Only carefully selected patients undergo lung resection, likely accounting for any discrepancies in survival rates. Some researchers have also reported prognostic factors, including the size of pulmonary metastases and number of resected pulmonary metastases. In the present study, we also identified size of lung metastasis as a prognostic factor, which is consistent with previous studies. Additionally, we identified pathologically-proven distant metastases at initial treatment of UC and lymph node metastases at lung surgery as prognostic factors. The large number of participants in the present study enabled identification of factors that only occur rarely.

This study had several limitations. First, it was a retrospective study using prospectively collected data from a registry database. Some measurement bias was inevitable due to the nature of the data collection. Furthermore, the database lacked information on several potential confounding factors, which, consequently, could not be included in our analysis. These omissions include “classic” pathologic features of primary tumors (such as histologic grade, mitotic figures, lymphovascular and perineural invasions, and histological variants), details about perioperative treatments (whether preoperative or postoperative chemotherapy or radiotherapy was administered), and the surgical approach for lung metastasectomy (open thoracotomy, video-assisted thoracoscopic surgery, or robot-assisted surgery). The absence of these data points represents a limitation in our study, potentially affecting the comprehensiveness of our analysis and the interpretation of our results. Second, the impact of surgery was not fully evaluated because we did not compare the outcomes of lung metastectomy with the outcomes of no surgery. However, lung resection has the advantage over other treatment modalities of enabling simultaneous pathological diagnosis and treatment of the lesion. It is sometimes difficult to distinguish pulmonary metastases from UC from primary lung cancer pathologically, necessitating sufficient specimens for pathological examination. For this reason, we recommend lung resection in suitable cases. Furthermore, this was a multi-institutional study, and each institution had autonomy on selection of study participants; some selection bias was, therefore, inevitable. Moreover, because of advances made during the study period of 30 years, such as positron emission tomography/CT and chemotherapy, including immunotherapy, neither preoperative diagnostic procedures nor treatment were uniform. Thus, not all study participants were diagnosed and treated equally. We cannot rule out the possibility that the results of this study strongly reflect the influence of some particular treatment. However, even with these limitations, we considered it useful to analyze the impact of pulmonary metastasectomy in a large patient cohort and identify prognostic factors to facilitate selection of treatment.

## 5. Conclusions

In conclusion, in this study we demonstrated that pulmonary metastasectomy for UC may play a curative role in highly selected patients and achieves much better OS and DFS than are achieved overall in patients with mUC. We believe that this is the largest study of patients who have undergone pulmonary resection of metastases from UC and that it provides benchmark data for future assessment of long-term outcomes of this rare entity.

## Figures and Tables

**Figure 1 cancers-16-03333-f001:**
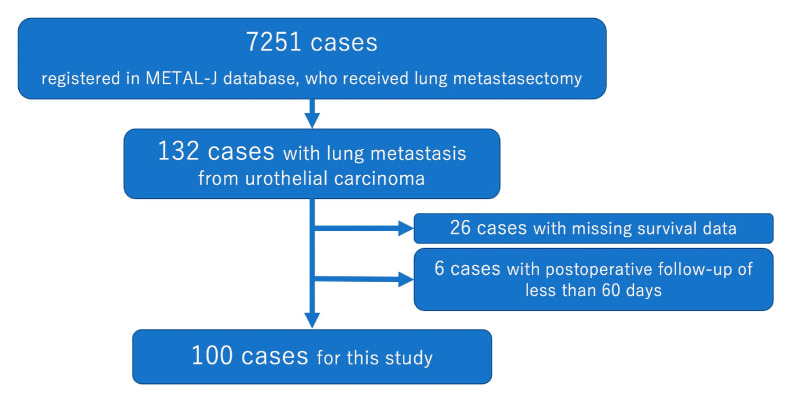
Flowchart showing patient selection.

**Figure 2 cancers-16-03333-f002:**
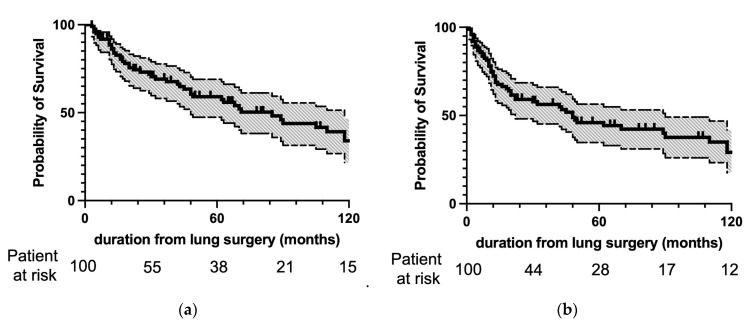
(**a**) Overall and (**b**) disease-free survival after pulmonary metastasectomy. The survival curves are depicted as solid lines, and 95% confidence intervals are in halftone. The numbers of patients at risk at lung surgery and 30, 60, 90, and 120 months after lung surgery are indicated on the abscissa. Three- and 5-year OS was 69% and 59%, respectively, and 3-year and 5-year DFS was 56% and 46%, respectively.

**Figure 3 cancers-16-03333-f003:**
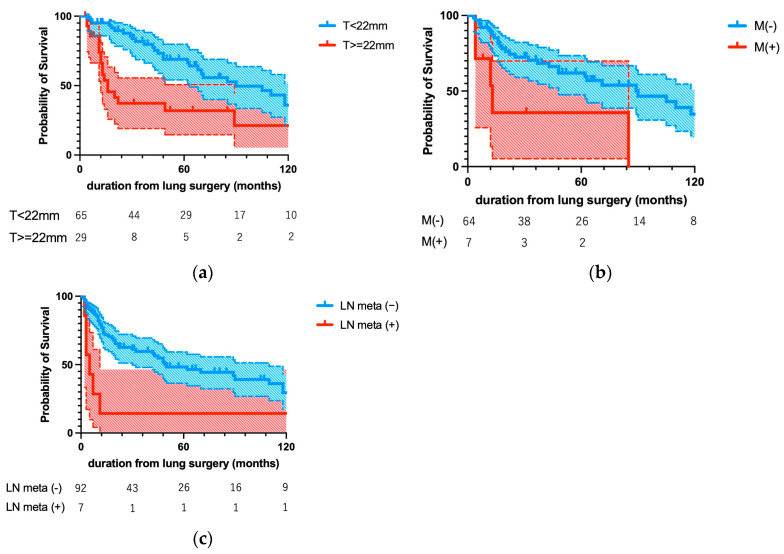
Kaplan–Meier curves for the prognostic factors identified as significant by multivariate analysis. (**a**) Distant metastases at the time of initial treatment of UC and (**b**) diameter of lung metastases > 22 mm were significantly associated with worse OS (*p* = 0.0061 and *p* = 0.0009, respectively), whereas (**c**) lymph node metastases at lung surgery were significantly associated with worse DFS (*p* = 0.0035).

**Table 1 cancers-16-03333-t001:** Patient characteristics.

Variable	Number/Mean ± SD (Range)
Sex	
Male	70
Female	30
Organ of primary tumor	
Bladder	63
Urinary tract	37
Initial treatment for primary tumor	
Surgery	44
Chemotherapy and/or radiotherapy	15
Unknown	55
Pathological N stage of primary tumor	
N0	62
N1–3	9
Pathological M stage of primary tumor	
M0	64
M1	7
Initial site of recurrence	
Pulmonary	95
Extrapulmonary	5
Number of tumors at detection of lung metastases	
Single	35
Multiple	11
Unavailable	54
Perioperative chemotherapy for lung metastases	
Before lung surgery	7
After lung surgery	11
None	84
Type of lung surgery	
Wedge resection	52
Segmentectomy	9
Lobectomy	14
Side of surgery for lung metastases	
Right	57
Left	35
Both	8
Lymph node metastases found at lung surgery	
Hilar lymph node metastases	2
Mediastinal lymph node metastases	6
Extended resection	
Chest wall	3
Angioplasty and bronchoplasty	1
Age at lung surgery (years)	67 ± 10 (26–84)
Size of lung metastases according to CT (mm)	30 ± 36 (1–172)
Size of resected lung metastases (mm)	21 ± 16 (5–86)
Disease-free interval from initial treatment (months)	26.6 ± 34.2 (0–172)
Interval from initial treatment to lung surgery (months)	30.4 ± 36.0 (0–172)
Follow-up period	54.6 ± 53.2 (3–224)

**Table 2 cancers-16-03333-t002:** Results of univariate analysis for overall and disease-free survival after lung metastasectomy.

Variables	Overall Survival		Disease Free Survival	
	HR (95% CI)	*p*-Value	HR (95% CI)	*p*-Value
Sex				
Male	Reference	0.95	Reference	0.6
Female	1.02 (0.54–1.92)		1.18 (0.65–2.14)	
Age at initial treatment of primary tumor	1.02 (0.98–1.06)	0.39	1.09 (0.43–2.79)	0.86
Organ of primary UC				
Bladder	Reference	0.51	Reference	0.34
Urinary tract	1.23 (0.66–2.28)		1.33 (0.74–2.39)	
Pathological N stage of primary UC				
N0	Reference	0.17	Reference	0.94
N1–3	1.24 (0.43–3.53)		1.04 (0.39–2.72)	
Pathological M stage of primary UC				
M0	Reference	0.006	Reference	0.19
M1	9.35 (1.89–46.2)		2.53 (0.63–10.2)	
Site of initial recurrence				
Pulmonary	Reference	0.42	Reference	0.29
Extrapulmonary	0.55 (0.13–2.38)		0.55 (0.19–1.64)	
Number of lung metastases detected by CT	1.02 (0.87–1.18)	0.82	1.39 (1.04–1.87)	0.028
Size of lung metastases according to CT	1.64 (1.25–2.16)	0.0004	2.18 (0.74–6.44)	0.16
Interval from treatment of primary tumor to detection of lung metastases	1.00 (0.99–1.01)	0.95	1.06 (0.94–1.19)	0.33
Age at lung surgery (years)	1.03 (1.00–1.06)	0.04	0.92 (0.37–2.32)	0.86
Number of resected lung metastases	1.07 (0.74–1.55)	0.71	1.52 (0.49–4.71)	0.47
Size of resected lung metastases	1.34 (1.12–1.60)	0.002	0.75 (0.24–2.30)	0.61
Interval from treatment of primary tumor to lung surgery	1.00 (0.99–1.01)	0.89	0.95 (0.82–1.10)	0.47
Type of lung surgery				
Wedge resection	Reference	0.12	Reference	0.62
Segmentectomy or lobectomy	1.78 (0.86–3.68)		1.18 (0.61–2.28)	
Lymph node metastases found at lung surgery				
Negative	Reference	0.0002	Reference	0.004
Positive	14.91 (3.54–62.8)		8.45 (2.02–35.4)	
Extended lung resection	1.73 (0.30–10.0)	0.54	1.29 (0.26–6.31)	0.75
Perioperative chemotherapy for lung metastases				
Before lung surgery	1.62 (0.48–5.55)	0.44	48.06 (8.11–284)	0.00004
After Lung surgery	1.24 (0.52–2.96)	0.63	1.02 (0.45–2.28)	0.97

**Table 3 cancers-16-03333-t003:** Results of multivariate analysis for overall and disease-free survival after lung metastasectomy.

Variables	Overall Survival		Disease Free Survival	
	HR (95% CI)	*p*-Value	HR (95% CI)	*p*-Value
**Pathological M stage of primary UC**				
M0	Reference	0.005		
M1	4.23 (1.54–11.6)			
**Number of lung metastases detected by CT**			1.12 (0.99–1.27)	0.08
**Age at lung surgery (years)**	1.02 (0.98–1.07)	0.32		
**Size of resected lung metastases**	1.62 (1.21–2.17)	0.001		
**Lymph node metastases found at lung surgery**				
Negative	Reference	0.26	Reference	0.012
Positive	1.99 (0.59–6.76)		7.68 (1.56–37.9)	
**Perioperative chemotherapy for lung metastases**				
Before lung surgery			1.40 (0.35–5.61)	0.64

## Data Availability

Data will be available on request due to restrictions due to privacy and ethical reasons.

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
