# Peer review of "Survival after Lung Metastasectomy from Urothelial Carcinoma: A Multi-Institutional Database Study"

_cancers, 2024, doi:10.3390/cancers16193333_

Round 1
Reviewer 1 Report
Comments and Suggestions for Authors
Urothelial carcinoma (UC) is a prevalent cancer, notably impacting the bladder and upper urinary tract. It stands as the 10th most common cancer globally and the second most common among genitourinary cancers. Despite advancements in treatment, metastatic urothelial carcinoma (mUC) presents a bad prognosis, with a 5-year overall survival rate below 5% for patients with metastatic disease. Systemic chemotherapy remains a main therspy, yet its effectiveness is limited, with many patients unable to receive it due to various health complications. This has led to an exploration of alternative treatment strategies, including the potential role of pulmonary metastasectomy for mUC patients.
The article provides a comprehensive analysis of pulmonary metastasectomy for mUC using data from the Metastatic Lung Tumor Study Group of Japan (METAL-J). The study identifies significant prognostic factors associated with long-term survival following pulmonary metastasectomy, such as the presence of pathologically-proven distant metastases at initial treatment, the size of lung metastases, and the presence of lymph node metastases. The findings contribute valuable insights into how surgical resection of lung metastases can potentially extend survival for selected patients, offering hope where conventional treatments fall short.
Despite its strengths, the study has notable limitations. Being retrospective, it could include biases related to data collection and patient selection. Key factors such as preoperative and postoperative chemotherapy, the type of lung surgery performed, and detailed pathological diagnostics were not uniformly reported or analyzed. Additionally, the study did not compare outcomes with non-surgical management, leaving a gap in understanding the relative benefit of metastasectomy.
Despite these limitations, the large cohort and detailed analysis make this study a significant contribution to the understanding of surgical options in the management of metastatic urothelial carcinoma.
I suggest adding the following scientific article links to the bibliography section for a more accurate representation of the references and the general topic of this study:
· 10.1159/000509434
- 10.1016/j.clgc.2021.12.005
Minor editing
Author Response
(comments)
I suggest adding the following scientific article links to the bibliography section for a more accurate representation of the references and the general topic of this study.
10.1159/000509434
10.1016/j.clgc.2021.12.005
(Answer)
Thank you very much for your valuable suggestions regarding additional references.
We appreciate your recommendation and have carefully considered both suggested articles. We have incorporated the article with DOI:10.1016/j.clgc.2021.12.005 into our bibliography, along with several other relevant references that further strengthen our literature review.
However, regarding the article with DOI:10.1159/000509434, we have decided not to include it in our bibliography. The reason for this is that the article focuses on metastatic prostate cancer, which is a different entity from the transitional cell carcinoma that is the main focus of our manuscript. While there may be some overlapping concepts in the management of metastatic disease, we believe that including this reference might potentially confuse readers or dilute the specificity of our literature review.
Reviewer 2 Report
Comments and Suggestions for Authors
Manuscript entitled "Survival after lung metastasectomy from urothelial carcinoma: A multi-institutional database study"
Major issues:
1. The authors are encouraged to include "classic" pathologic features of primary tumors and metastatic tumors into analysis (including survival analysis), which includes: histologic grade, mitotitc figures, lymphovascular and perineurial invasions, and histological variants (Ref.
Modern Pathology Volume 22, Supplement 2, June 2009, Pages S96-S118).
2. The resection margin status (R0, R1, ... etc.) should also be included into analysis.
3. The use of chemo- or targeted therapy before resection should also be included into analysis.
4. The images (radiographic and pathology) of representative cases should be shown.
Comments on the Quality of English LanguageAcceptable.
Author Response
Comment 1. The authors are encouraged to include "classic" pathologic features of primary tumors and metastatic tumors into analysis (including survival analysis), which includes: histologic grade, mitotitc figures, lymphovascular and perineurial invasions, and histological variants (Ref. Modern Pathology Volume 22, Supplement 2, June 2009, Pages S96-S118).
Answer 1. Thank you for your valuable comments regarding our manuscript.
Our study is based on a database that, unfortunately, does not contain detailed pathologic features of primary tumors as you've mentioned. We acknowledge that this is a significant limitation of our research. The absence of data on histologic grade, mitotic figures, lymphovascular and perineural invasions, and histological variants indeed restricts our ability to perform more comprehensive analyses, including survival analysis based on these "classic" pathologic features.
We have addressed this limitation in the discussion section of our paper, highlighting the potential impact on our findings and suggesting it as an area for future research. This gap in our data underscores the importance of more detailed pathological reporting in database studies to enable more nuanced analyses in future investigations.
Comment 2. The resection margin status (R0, R1, ... etc.) should also be included into analysis.
Answer 2. In the treatment of the primary tumor, for cases with surgery in which the resection margin could be determined, all were complete resections. On the other hand, in the treatment of lung metastases, there were two cases where tumors remained on the operative side after lung surgery. However, due to the small number of cases, further analysis was difficult. This point has been added to the Results section.
Comment 3. The use of chemo- or targeted therapy before resection should also be included into analysis.
Answer 3. We have indeed included information about preoperative therapy in our analysis, as reflected in Tables 1, 2, and 3 under the factor "Perioperative chemotherapy for lung metastases." These tables show the number of cases that received treatment "before lung surgery."
Specifically, we identified 7 cases that received preoperative treatment. The chemotherapy regimens for these cases were as follows:
- DD-MVAC therapy: 2 cases
- MVAC therapy: 2 cases
- GEM+CDDP: 1 case
- GEM+CBDCA: 1 case
- CBDCA+TXL: 1 case
Given the diversity of regimens and the small number of cases for each, it was challenging to perform a meaningful analysis based on individual regimen types. Therefore, we opted to analyze these cases collectively under the category of "before lung surgery." This point has been added to the Results section.
Comment 4. The images (radiographic and pathology) of representative cases should be shown.
Answer 4. Unfortunately, we are unable to provide images of representative cases. This study is based on database information from which all personal identifiers were removed. As a result, we cannot identify specific patients or access their individual radiographic or pathological images. The anonymization of the database, while essential for protecting patient privacy, prevents us from retrieving such visual data for presentation.
Round 2
Reviewer 1 Report
Comments and Suggestions for Authors
Authors answered all comments and suggestions.
Comments on the Quality of English LanguageMinor editing